# Mitochondrial lipidomes are tissue specific – low cholesterol contents relate to UCP1 activity

Sarah Brunner[1,2,*], Marcus Höring[1,*], Gerhard Liebisch[1], Sabine Schweizer[2] , Josef Scheiber[3], Piero Giansanti[4] , Maria Hidrobo[2] , Sven Hermeling[1,2] , Josef Oeckl[5], Natalia Prudente de Mello[6,7], Fabiana Perocchi[6,8,9] , Claudine Seeliger[2], Akim Strohmeyer[5] , Martin Klingenspor[5] , Johannes Plagge[2], Bernhard Küster[10,11], Ralph Burkhardt[1] , Klaus-Peter Janssen[12], Josef Ecker[1,2]

**Lipid composition is conserved within sub-cellular compartments to maintain cell function. Lipidomic analyses of liver, muscle, white and brown adipose tissue (BAT) mitochondria revealed substantial differences in their glycerophospholipid (GPL) and free cholesterol (FC) contents. The GPL to FC ratio was 50-fold higher in brown than white adipose tissue mitochondria. Their purity was verified by comparison of proteomes with ER and mitochondria-associated membranes. A lipid signature containing PC and FC, calculated from the lipidomic profiles, allowed differentiation of mitochondria from BAT of mice housed at different temperatures. Elevating FC in BAT mitochondria prevented uncoupling protein (UCP) 1 function, whereas increasing GPL boosted it. Similarly, *STARD3* overexpression facilitating mitochondrial FC import inhibited UCP1 function in primary brown adipocytes, whereas a knockdown promoted it. We conclude that the mitochondrial GPL/FC ratio is key for BAT function and propose that targeting it might be a promising strategy to promote UCP1 activity.**

## Introduction

Cells invest substantial resources to generate an enormous repertoire of lipid species with diverse physico-chemical and structural properties. Lipid species composition is organ-specific, cell type–specific and even conserved at the sub-cellular level (Ecker et al, 2010; Schweizer et al, 2019; Ecker et al, 2021). The plasma membrane's lipid composition, for example, drastically differs from those of mitochondria (van Meer et al, 2008). Primarily occurring lipid classes are glycerophospholipids (GPL) such as phosphatidylcholine (PC) and sterol-free cholesterol (FC). The GPL/FC ratio is a critical parameter controlling lipid packing density in membranes because cholesterol intercalates between phospholipid chains (van Meer et al, 2008; Ernst et al, 2016). Its planar structure promotes the transition from fluid to solid gel phases in membranes and increases viscosity (Ernst et al, 2016). In organelles, GPL/FC ratios typically range from 1.0 (50% FC) for the plasma membrane to 9.0 (10% FC) for mitochondria (van Meer et al, 2008). Because lipid bilayers are dynamic, imbalances in lipid composition, including the GPL/FC ratio, leading to altered membrane biophysics, are equalized to ensure cell functionality (Ernst et al, 2016; Levental et al, 2020). Though, substantial interferences such as lipid overload triggering the metabolic syndrome cannot be compensated.

Until now, it has been largely unclear whether organelle lipid architectures systematically differ between organs, although implied by the fundamental biological maxim "structure subserves function." Mitochondria are the provider of cellular energy from aerobic respiration in all tissues. Those present in brown adipose tissue (BAT) contain immense amounts (~8% of total mitochondrial protein) of uncoupling protein (UCP) 1, which enables them to efficiently convert chemical energy stored as triglycerides into heat; a process called non-shivering thermogenesis (Lin & Klingenberg, 1980; Rousset et al, 2004). UCP1 creates a proton leak in the inner mitochondrial membrane (IMM), which dissipates the proton motive force. It is induced by free fatty acids, which are released after stimulation with beta-adrenergic agonists, such as norepinephrine generated at cold exposure (Cannon & Nedergaard, 2004), and inhibited by purine nucleotides such as GDP (Porter, 2017). To ask if

[1]Institute of Clinical Chemistry and Laboratory Medicine, University Hospital Regensburg, Regensburg, Germany   [2]ZIEL Institute for Food & Health, Research Group Lipid Metabolism, Technical University of Munich, Freising, Germany   [3]BioVariance GmbH, Waldsassen, Germany   [4]Bavarian Center for Biomolecular Mass Spectrometry at the University Hospital rechts der Isar, Technical University of Munich, Munich, Germany   [5]Chair of Molecular Nutritional Medicine, TUM School of Life Sciences, Technical University of Munich, Freising, Germany   [6]Institute for Diabetes and Obesity, Helmholtz Diabetes Center (HDC), Helmholtz Zentrum München and German National Diabetes Center (DZD), Neuherberg, Germany   [7]Graduate School of Systemic Neurosciences (GSN), Ludwig-Maximilians University, Munich, Germany   [8]Institute of Neuronal Cell Biology, Technical University of Munich, Munich, Germany   [9]Munich Cluster of Systems Neurology, Munich, Germany   [10]Chair of Proteomics and Bioanalytics, Technical University of Munich, Freising, Germany   [11]Bavarian Biomolecular Mass Spectrometry Center, Technical University of Munich, Freising, Germany   [12]Department of Surgery, School of Medicine, University Hospital rechts der Isar, Technical University of Munich, Munich Germany

Correspondence: josef.ecker@ukr.de
*Sarah Brunner and Marcus Höring contributed equally to this work

cellular respiration or thermogenic function requires a specific lipidomic environment, we applied the following experimental strategy:

(I) A practicable mitochondria isolation procedure was established using murine BAT and liver. Enrichment and purity were tested by a full proteome analysis and immunoblotting of marker proteins. (II) The lipidome of mitochondria isolated from BAT, liver, skeletal muscle and white adipose tissue (WAT) was quantified, unveiling that BAT mitochondria contain almost exclusively GPL, but little or no cholesterol, leading to a GPL/FC ratio of ~50 (2% FC). (III) To strengthen these findings, machine learning was applied as an independent approach for data analysis. We found that mitochondrial PC and FC contents allow prediction of their tissue origin for unknown samples and that the GPL/FC ratio can be associated with adipose tissue browning. (IV) To prove the physiological relevance of the BAT-specific lipidomic environment, UCP1 activity, relevant for thermogenic function, was analyzed in isolated BAT mitochondria incubated with donor vesicle preparations containing different amounts of PC and FC; and in primary brown adipocytes, in which steroidogenic acute regulatory protein (STARD) 3 (also known as metastatic lymph node 64 protein, MLN64), facilitating mitochondrial FC import (Elustondo et al, 2017), was manipulated.

# Results

## Mitochondria isolated by differential centrifugation are highly enriched containing negligible fractions of ER and mitochondria-associated membranes (MAM)

Mitochondrial fractions were isolated from murine BAT and liver tissue by differential centrifugation, referred to as crude mitochondria (cMito). Mitochondria and ER are connected via junctions termed MAM (Kornmann et al, 2009; Kornmann, 2013), making them potential co-isolates of mitochondria. Therefore, the purity of cMito was checked by comparing its full proteome with ER, MAM and mitochondrial fractions isolated by density gradient centrifugation, referred to as pure mitochondria (pMito). In total, 7,149 proteins were identified with the highest numbers in the MAM and ER fractions of the liver (Figs 1A, B, I, and J and S1A–C and H–J). 77% (BAT) and 91% (liver) of proteins detected in MAM were identical to those found in cMito; 81% (BAT) and 89% (liver) of ER proteins to those found cMito (Fig 1A and I). 247 (BAT) and 114 (liver) of the 1,000 most abundant ER proteins (Figs 1C and K and S1D and K), 246 (BAT) and 21 (liver) of the 1,000 most abundant MAM proteins (Figs 1D and L and S1E and L) were not detected or detected at low levels in the cMito fraction, suggesting very low contaminations of cMito with these organelles. These included several proteins localized to the ER or MAM (Fig S1D, E, K, and L, red dots), such as SCD1 and MAP3K5 (Nishitoh et al, 2002; Man et al, 2006; Xiang et al, 2017). pMito comprised fewer proteins than cMito (Fig S1C and J). However, the distribution of expression levels of the 1,000 most abundant proteins (after classification into not expressed [NE], low, medium, and high) was more similar between cMito and pMito than with ER or MAM (Fig 1E and M). The signal intensities of uniquely occurring proteins highly correlated between pMito and cMito preparations

with $R^2 > 0.76$, (Fig 1F and N) and were more comparable to those of MAM than to ER (Figs 1G and O and S1G and N).

## Mitochondrial lipidomes, particularly their phosphatidylcholine and free cholesterol contents, are tissue specific

To test whether the lipidomic composition of liver and BAT mitochondria differs, a comprehensive quantitative lipidomic analysis was performed using high-resolution mass spectrometry (HR-MS) and tandem mass spectrometry (QQQ), including glycerophospholipids (GPL: PC, phosphatidylcholine; PE, phosphatidylethanolamine; PE P, PE-based plasmalogens; PI, phosphatidylinositol; PS, phosphatidylserine; cardiolipin, CL), sphingolipids (SL: SM, sphingomyelin; Cer, ceramide; Hex-Cer, hexosylceramide), glycerolipids (GL: DG, diglycerides; TG, triglycerides), and sterols (ST: FC, free cholesterol; CE, cholesteryl ester). In total, 275 lipid species were quantified. The lipidomic composition of pMito and cMito was very similar ($R^2 > 0.97$, rho > 0.83) (Fig 1H and P). Moreover, the isolation of cMito requires far fewer animals (five mice per sample) than for pMito preparations (15 mice per sample) and is less time-consuming, which is relevant considering the respiratory capacity of mitochondria. For these reasons, mitochondria isolated by differential centrifugation (cMito) were used for further investigations.

A 2.4-fold higher total lipid content (related to protein) was found in mitochondria isolated from liver compared to BAT, and between 85% (liver) and 90% (BAT) were GPL (Fig 2A). PC, PI, PS levels were lower in BAT, as well as the SL (0.8–2.6% of total lipids) Cer, Hex-Cer, and SM. CL (3–10% of total lipids), TG and DG concentrations were similar. Surprisingly, BAT mitochondria contained almost no FC. The levels of 39 GPL lipid species were significantly different between the two groups (Fig 2D). BAT comprised less GPL species with highly unsaturated acyl chains (>3 DB), including PC 38: 5, PC38:6, PE38:6, PI 38:5, PI38:6, PI40:5, PS 40:5 (Figs 2D and S2A–L).

To ask whether these lipid profiles are specific for BAT mitochondria and to validate our findings, mitochondria isolated from a new set of BAT samples were analyzed and compared with epididymal WAT (eWAT) and muscle. Here, we focused on the differentially regulated lipid classes FC, GPL (PC, PE, PI, PS), and SL (Cer, Hex-Cer, SM). In contrast to liver samples, GPL levels, that is, PC and PE, were lower (Fig 2B and C) but more unsaturated (more DB > 5) (Figs 2E and F and S3A–J and S4A–J) in eWAT and muscle compared to BAT. Whereas FC contents of muscle mitochondria were in the range between liver and BAT, those purified from eWAT contained sixfold higher amounts of FC (equivalent to 37% of all analyzed lipids) than BAT (Fig 2B and C). A Western blot analysis of ER, MAM, and mitochondrial marker proteins suggests that WAT mitochondria have the same purity as those isolated from BAT (Fig S5A–C). Furthermore, it shows that the plasma membrane, which is the FC-richest organelle (van Meer et al, 2008) (Table S1), contaminates WAT and BAT isolates to similar degrees (Fig S5D).

Together, mitochondria isolated from various tissues differed significantly in their lipidomic composition. Those purified from liver and eWAT had significantly higher amounts of FC, whereas total amounts of the major GPL PC and PE were lower in eWAT and muscle samples compared to BAT. The GPL/FC, PC/FC, and PE/FC ratios were almost 50-fold higher in BAT mitochondria than in those of eWAT (Fig 2G–I).

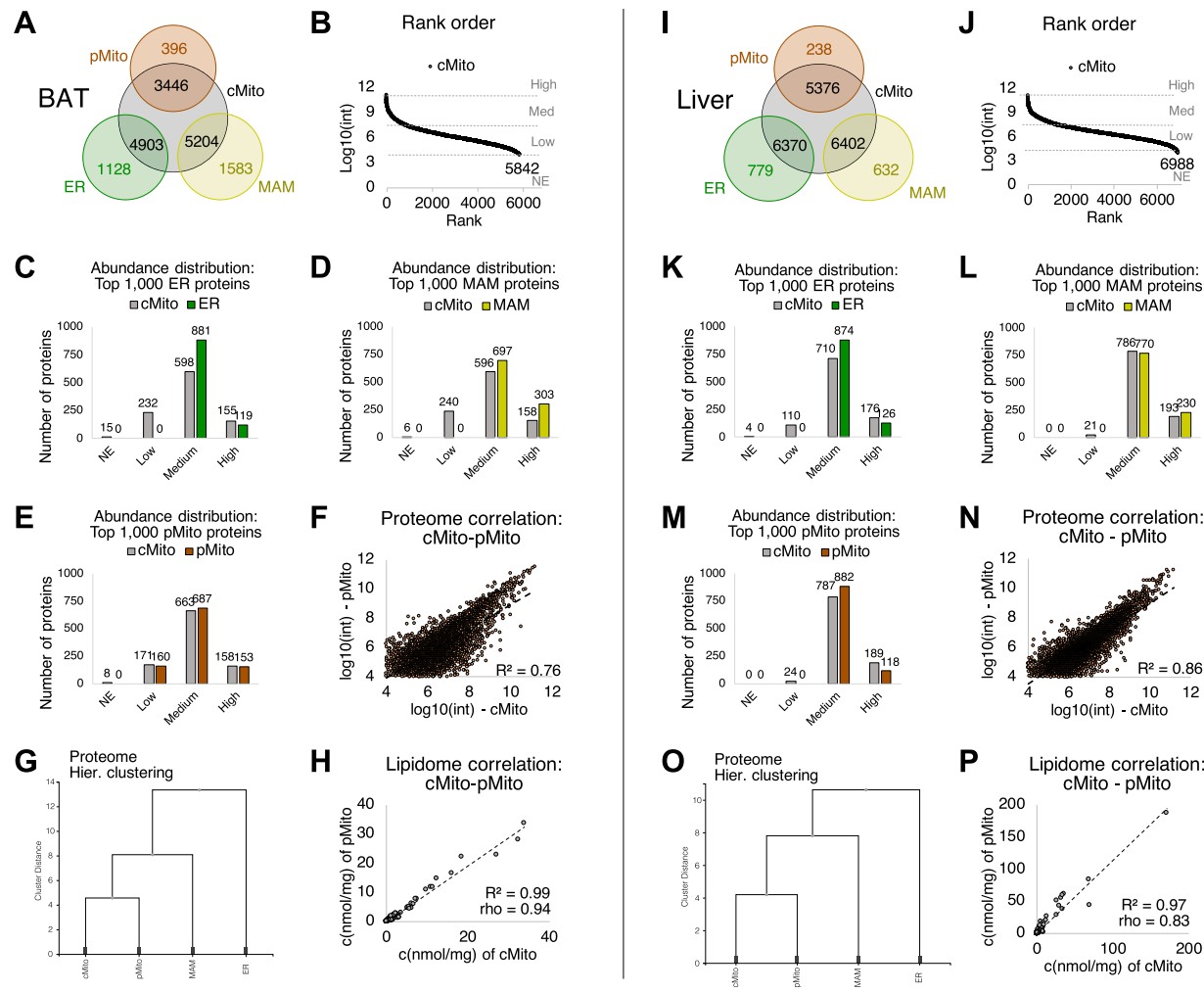

**Figure 1.   Mitochondria isolated by differential centrifugation are highly enriched and contain negligible fractions of ER and MAM.**
**(A, B, C, D, E, F, G, H, I, J, K, L, M, N, O, P)** Data from BAT organelles are shown in (A, B, C, D, E, F, G, H), those from liver are shown in (I, J, K, L, M, N, O, P). **(A, I)** Venn diagrams of proteins detected in cMito and pMito or ER or MAM. **(B, J)** Rank order of proteins after a full proteome analysis for cMito. **(C, D, E, K, L, M)** Distribution of the top 1,000 ranked proteins according to their signal intensities in ER (C, K), MAM (D, L), and pMito (E, M) in comparison with cMito after classification into not expressed (NE; $\log_{10}$ [int] < 4), low ($\log_{10}$ [int] 4–6), medium ($\log_{10}$ [int] 7–9), and high ($\log_{10}$ [int] > 9) expressed proteins. **(F, N)** Correlation of signal intensities from all proteins detected in pMito and cMito. **(G, O)** Hierarchical cluster analysis of the 1,000 most abundant pMito proteins compared with cMito, ER, and MAM. All proteomic data were obtained from ER, MAM, cMito, and pMito samples prepared from pooled BAT of 5–15 mice or liver of five mice. **(H, P)** Lipidome correlation between lipid species quantified in pMito (isolated from pooled BAT of 15 mice or liver of five mice) and cMito (means of n = 7 preparations with each isolated from liver samples or pooled BAT of five mice) after a lipidomic analysis. $R^2$ indicates Pearson's, rho indicates Spearman's correlation coefficients.
Source data are available for this figure.

### A BAT-specific lipid signature comprising phosphatidylcholine and cholesterol predicts the tissue origin of mitochondria and can be related to adipose tissue browning

To further strengthen our findings and to reduce our large set of lipids to candidates specifically enriched in BAT, we applied machine learning. Therefore, we ran three models based on Random Forest (RF) allowing a precise classification of observations (= lipid species) into categories (= tissue origin of mitochondrial samples). RF is a supervised decision tree learning procedure that can handle large numbers of variables at small sample sizes without overfitting (Breiman, 2001; Biau, 2012). First, a RF model was applied using GPL species (PC, PC O, PE, PE P, PG, PI, PS) and FC (in total: 94 lipid

species) obtained from a subset of liver, muscle, BAT, and eWAT samples as input for variable reduction and selection. The top three lipid species most effectively classifying our samples were (1) PC 38: 2, (2) FC, (3) PE P-16:0/22:6 (Table S2). Next, we trained two models that differed by using either the top (A) three or (B) two lipid species for sample classification. Classification based on both models was not only effective on our samples from the training set with high sensitivity and specificity (receiver operator characteristic - area under the curve: (A) 0.932, (B) 0.927), but also on an independent sample set consisting of BAT, WAT, and liver samples for verification (Fig 3A–D). Both models performed similarly, with a slightly higher verification Receiver operator characteristic area under the curve for the model with two variables (0.856 versus 0.809). All three

**Figure 2. The mitochondrial lipidome including GPL and FC of BAT mitochondria significantly differs to those of the liver, WAT, and muscle.**

**(A)** Lipid class composition of mitochondria from BAT (n = 7) versus liver (n = 4). **(B)** BAT (n = 3) versus eWAT (n = 3). **(C)** BAT (n = 3) versus skeletal muscle (n = 3). Volcano plots show significant differences (−Log₁₀ [*P*-value]) of log₂ fold change in lipid classes. Orange dots indicate lipid species significantly decreased in BAT and blue dots indicate lipid species significantly increased in BAT; Stacked barplots show total lipid levels and distribution into SL, GL, Sterols, and GPL (means); Barplots show lipid class compositions (means +SD), *P < 0.01 after FDR correction. **(D)** GPL species and FC composition of mitochondria from BAT (n = 7) versus liver (n = 4). **(E)** BAT (n = 3) versus eWAT (n = 3). **(F)** BAT (n = 3) versus skeletal muscle (n = 3); Volcano plots show significant differences (−Log₁₀ [*P*-value]) of log₂ fold change in GPL species and FC. Each dot represents a specific lipid species and the color indicates to which lipid class it belongs; Stacked barplots show distribution of saturation in GPL as number of double bonds (means); Barplots show PC species composition (means ± SD), *P < 0.01 after FDR correction. **(G)** GPL/FC. **(H)** PC/FC. **(I)** PE/FC. Calculated from all analyzed mitochondria samples of BAT (n = 10), liver (n = 4), muscle (n = 3) and eWAT (n = 3); GPL included PC, PC O, PE, PE P, PS, PI, PG; Shown are means ± SD, *P < 0.01 indicates a significant difference relative to BAT, determined using a two-sided *t* test. Each mitochondrial sample was isolated from tissue pooled of n = 5 mice.

Source data are available for this figure.

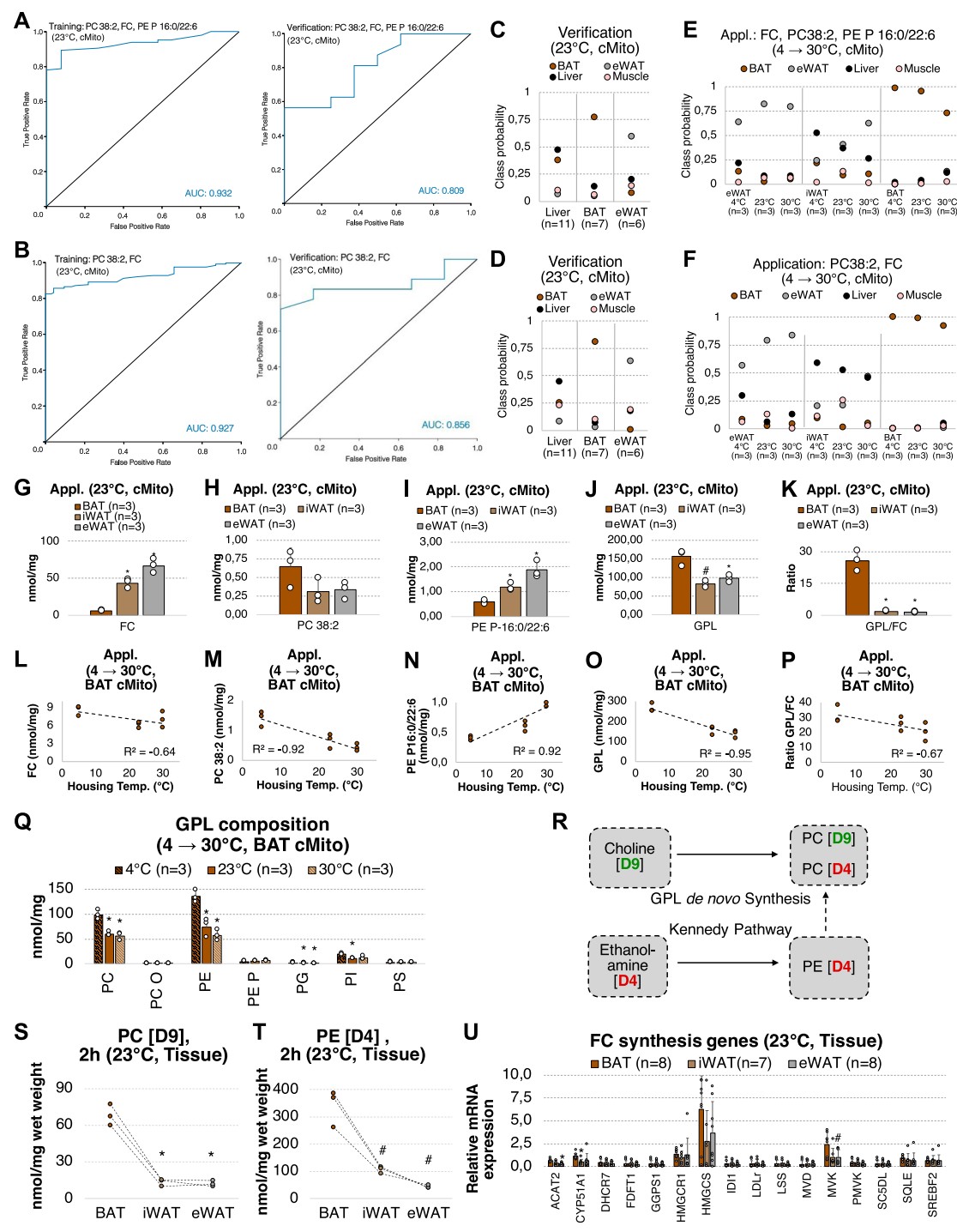

**Figure 3. A BAT-specific lipid signature comprising PC38:2 and FC as well as the GPL/FC ratio can be related to adipose tissue browning.**
**(A, B)** ROC curves of prediction models based on Random Forest (RF) containing either three lipid species (PC38:2, FC, PE P 16:0/22/6) (A) or 2 lipid species (PC38:2, FC) (B) for training (BAT, n = 6; liver, n = 3; eWAT, n = 3; muscle, n = 3) and verification (liver, n = 11; BAT, n = 7; eWAT n = 6; muscle was not tested) sample sets. **(C, D)** Classification probabilities, which indicate the individual prediction confidences for BAT, eWAT, the liver and muscle, of both models tested on the verification sample sets. A score of 1 equals to 100% confidence. Shown are the means. **(E, F)** Classification probabilities of the models tested on a new set of mitochondria isolated from eWAT, iWAT, and BAT of mice housed at 4°C, 23°C, and 30°C of n = 3 per condition. Shown are means. iWAT was not part of the training sample set and, hence, cannot be predicted as such. g, FC, h, PC 38:2, i, PE P 16:0/22:6, j, GPL, k, GPL/FC ratio in BAT, iWAT and eWAT mitochondria from mice housed at 23°C. **(E, F)** Samples are identical to those used in (E, F); Shown are means of n = 3 ± SD. (L, M, N, O, P) Correlation of FC, (m) PC 38:2, (N) PE P 16:0/22:6, (O) GPL, and (P) GPL/FC from BAT mitochondria with mice housing temperatures. R² indicate Pearson's correlation coefficients. **(Q)** GPL composition of BAT mitochondria from mice housed at 4°C, 23°C, and 30°C. Shown are means ± SD of three mice. **(R)** Labeling strategy to quantify de novo PC and PE synthesis in mice. PE (D4) to PC (D4) conversion was not detected in adipose tissue. **(S, T)** PC (D9), (T) PE (D4) levels detected 2 h after i.p. injection of choline (D9) and ethanolamine (D4) in BAT, iWAT, and eWAT of n = 3 mice housed at 23°C. Associated samples originating from the identical mouse are linked by dashed lines. **(U)** mRNA expression of cholesterol synthesis genes in BAT (n = 8), iWAT (n = 7), and eWAT (n = 8) from mice housed at 23°C. Shown are

sample types (BAT, eWAT, liver; muscle data were not available for verification) could be clearly classified, although classification probabilities were higher for BAT and WAT mitochondrial preparations than for liver (Fig 3C and D).

Finally, both RF models were tested on lipidomic data obtained from mitochondria purified from adipose tissue of mice housed at 4°C, 23°C or 30°C for 1 wk to either promote or antagonize adipose tissue browning. In addition to eWAT and BAT, mitochondria were also isolated from inguinal WAT (iWAT) containing white and brite (also called beige) adipocytes with a mixed phenotype between white and brown adipocytes (Wu et al, 2012; Rosenwald et al, 2013). The browning capacity of iWAT is lower as it is in BAT, but higher as it is in eWAT. At 30°C, the prediction probabilities of all analyzed samples for BAT decreased, whereas at 4°C the prediction probabilities for WAT decreased, with both models (Fig 3E and F). Mitochondria purified from iWAT were not part of the training sample set and, hence, cannot be predicted as such. They were classified as originating from liver (Model B, two lipid species), or either liver or eWAT (Model A, three lipid species) depending on the housing temperature. With rising temperatures from 4°C to 30°C, the classification probabilities of iWAT to be eWAT increased, and to be BAT decreased, respectively. These findings confirm that the different mitochondria types most significantly differentiate in their FC and PC contents. Furthermore, they show that these two lipids are sufficient to predict the tissue origin of mitochondrial preparations from eWAT and BAT.

Mitochondrial FC levels differed dramatically between all three adipose tissue depots of mice housed at room temperature (23°C) (Fig 3G and I). The abundance of PC 38:2 and the total GPL content were ~twofold elevated in BAT mitochondria compared with WAT (Fig 3H and J) and correlated with decreasing mice housing temperatures (Fig 3M and O), adipose tissue browning, respectively. The concentration of PE P 16:0/22:6 (only included in model A) negatively correlated with adipose tissue browning capacity (Fig 3I and N). Most importantly, the GPL/FC ratio was ~20-fold higher in mitochondria from BAT than from iWAT or eWAT (Fig 3K and P), and total GPL levels could be associated with adipose tissue browning (Fig 3O).

### GPL de novo synthesis capacity is higher in brown than in WAT

PC and PE made up 86% of total GPL and were elevated in BAT mitochondria originating from mice housed at 4°C compared with 23°C or 30°C (Fig 3Q). Thus, we next asked whether BAT and WAT have differential de novo synthesis capacities for PC and PE. Therefore, mice (housed at 23°C) were intraperitoneally (i.p.) injected with choline (D9) and ethanolamine (D4) for 2 h before quantifying their incorporation into GPL in adipose tissue (Fig 3R). We found threefold higher PC (D9) and PE (D4) levels in BAT compared to the eWAT and iWAT (Fig 3S and T), suggesting that the high GPL/FC ratio determined in BAT is driven by a higher de novo PC and PE synthesis. We conclude that cholesterol synthesis is

similar in WAT and BAT, as the mRNA expression of related enzymes was comparable between these tissues (Fig 3U).

### G3P-dependent respiration and UCP1 function in BAT mitochondria depend on their GPL/FC ratio

The virtual absence of free cholesterol in BAT mitochondria and the association of the GPL/FC ratio with adipose tissue browning suggest a causal role for thermogenic function, including uncoupled respiration mediated by UCP1. We therefore tested whether the GPL/FC ratio controls the UCP1 function. Intact mitochondria purified from BAT of WT mice (referred to as "WT mitochondria") were incubated with small unilamellar donor vesicles (DV) composed of varying ratios of GPL and FC (95% PC, 5% phosphatidic acid [PA]; 75% PC, 5% PA, 20% FC; 55% PC, 5% PA, 40% FC; 35% PC, 5% PA, 60% FC) in the presence of methyl-α-cyclodextrin. PA with a conical shape and a very small headgroup was added to donor vesicles to decrease their size and fasten their fusion with mitochondria (Hauser & Gains, 1982; Clark et al, 2020). As illustrated in Fig 4A, DV are hydrophilic on their outer shell exposed to the mitochondria, whereas their inner core is hydrophobic. Methyl-α-cyclodextrin is used as a carrier facilitating fusion of GPL-containing vesicles with lipid bilayers (Kainu et al, 2010; Li et al, 2016). To the current knowledge, translocation of lipids from the outer to the IMMs is protein-assisted. This process likely involves steroidogenic acute regulatory proteins (STARD) such as STARD 1 for FC and PRELI proteins for GPL (Miller, 2007; Tamura et al, 2020). A potential for contamination by the vesicles used to treat the mitochondria is highly unlikely but still possible.

The following mitochondrial bioenergetics profile was measured using microplate-based respirometry (Fig 4B): (1) basal respiration; (2) inhibition of ATP synthase by oligomycin (Oligo) to distinguish oxygen consumption used for ATP synthesis (coupled respiration) from the basal proton leak (basal uncoupled respiration); (3) fueling general respiration with glycerol-3-phosphate (G3P); (4) inhibition of UCP1-mediated respiration with guanosine diphosphate (GDP); (5) blocking complex III of the electron transport chain with antimycin A (Anti A) to inhibit mitochondrial respiration and leave only non-mitochondrial oxygen consumption. UCP1 activity was calculated by subtraction of the $O_2$ consumption rates (OCR) at (4) (UCP1 inhibition, GDP) from the OCR at (3) (general respiration and UCP1 fueling).

We could clearly link G3P-dependent respiration and UCP1 activity with mitochondrial PC and FC contents. Donor vesicles containing solely GPL increased the mitochondrial GPL/FC 3.5-fold, PC/FC 8.1-fold (Fig 4H) and UCP1 activity 1.7-fold compared to untreated WT mitochondria (Fig 4C and F). Addition of 20–60% FC to donor vesicles in turn stepwise elevated the total mitochondrial FC content, reduced the PC fraction (Fig 4G), the GPL/FC (Fig 4H) and, most importantly, the UCP1 function (Fig 4F). The FC and GPL contents added to the donor vesicles (Fig 4I and J) and those

---

means ± SD. **(G, H, I, J, K, Q, S, T)** *$P < 0.01$, #$P < 0.05$ indicate a significant difference relative to BAT 23°C (cMito, (G, H, I, J, K)), BAT 4°C (cMito, (Q)), or BAT 23°C (S, T); determined using a two-sided $t$ test. Each mitochondrial sample was isolated from tissue pooled of n = 5 mice.
Source data are available for this figure.

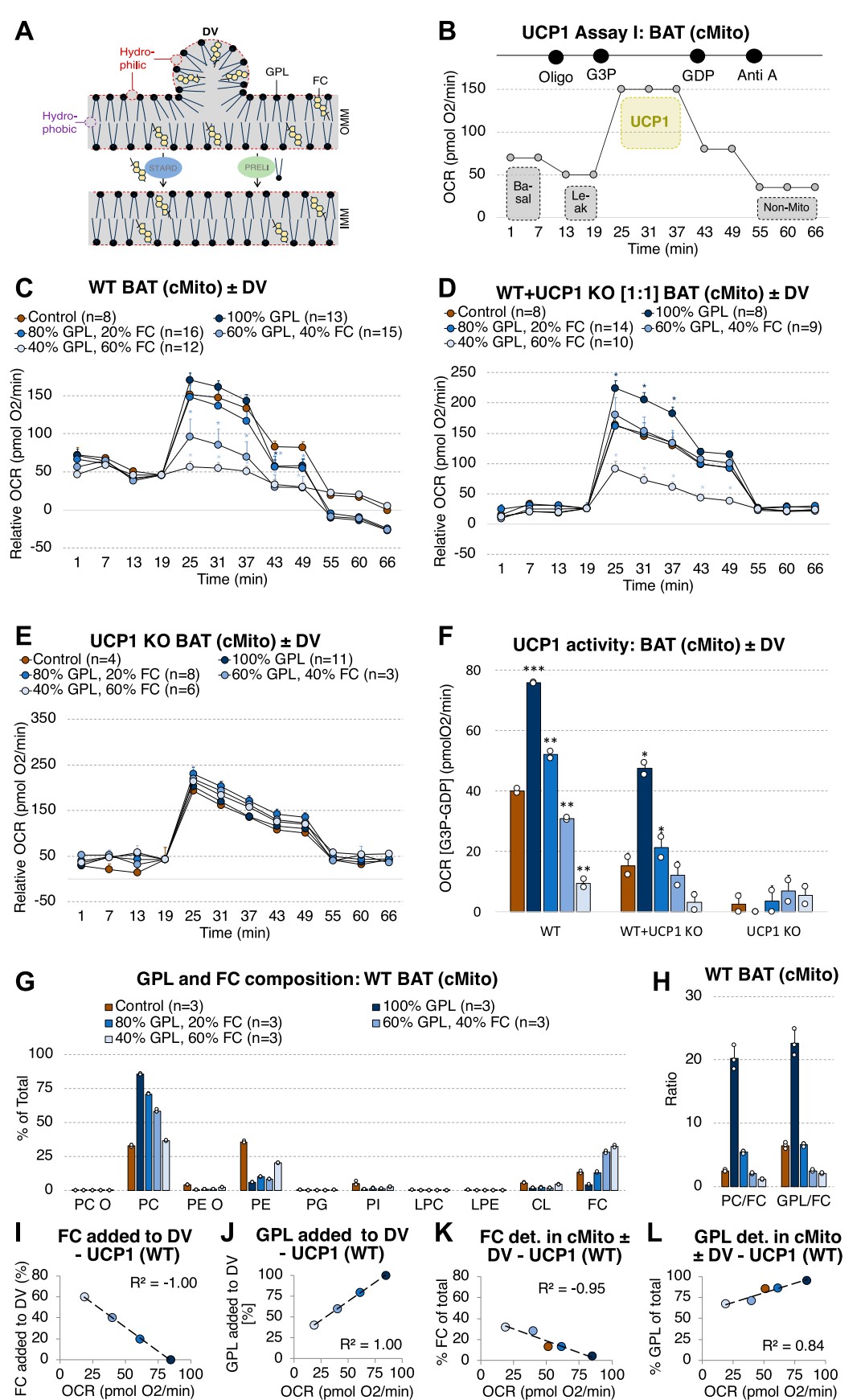

measured in mitochondria by mass spectrometry after treatment with donor vesicles (Fig 4K and L) correlated highly with UCP1 activity with $R^2$ of 0.84–1.0.

To strengthen these findings and to validate that our respiratory assay allows an estimation of UCP1 activity, we isolated mitochondria from BAT of UCP1 KO mice. They were either analyzed directly (referred to as "UCP1 KO mitochondria") or mixed with WT mitochondria in a ratio of 1:1 (referred to as "mixed WT-UCP1 KO mitochondria"). Whereas in mixed WT-UCP1 KO mitochondria UCP1 activity was approximately half as high as in WT mitochondria, in UCP1 KO mitochondria UCP1 function was almost absent (Fig 4F). In WT mitochondria, the measured OCRs after G3P injection (25-31-37 min) can be related by linear regression. After GDP injection, at 43 min, the OCR strongly drops because of inhibition of UCP1-mediated respiration and reduced $O_2$ consumption (Fig S6A). By contrast, in UCP1 KO mitochondria, GDP has no effects on G3P-induced $O_2$ consumption. The OCRs measured after G3P injection (25-31-37 min) can be related to the OCR determined after GDP injection (43 min) (Fig S6B), indicating that UCP1 activity is not present.

Incubation of mixed WT-UCP1 KO mitochondria with DV comprising only GPL significantly increased UCP1 activity. Addition of FC reduced it again (Fig 4D and F), as observed for WT mitochondria. Treatment of UCP1 KO mitochondria with DV did neither significantly nor systematically alter G3P-mediated respiration and UCP1 activity (Fig 4E and F).

The choice of substrates and their sequence of application can influence respiratory analyses in BAT mitochondria including UCP1 activity, as shown by the Nedergaard & Cannon group (Shabalina et al, 2010). To test whether ATPase inhibition before injection of G3P might bias our results, UCP1 activity of WT mitochondria incubated with selected DV was analyzed in the absence of oligomycin (Fig S7A). In accordance with our previous results, DV containing solely GPL elevates UCP1 activity, whereas the addition of 20% FC lowers it (Fig S7B and C).

To test if the GPL/FC impacts the maximal capacity of the respiratory chain, respiration was measured in isolated mitochondria treated with GDP (to inhibit UCP1) and carbonyl cyanide 4-(trifluoromethoxy) phenylhydrazone (FCCP) as uncoupling agent using high-resolution and microplate-based respirometry. With both methods we could not detect any significant differences in maximal respiratory capacity between mitochondria without or incubated with DV containing different GPL and FC contents (Fig S7D and E).

Finally, to investigate whether UCP1-independent respiration, including basal, basal uncoupled and non-mitochondrial respiration, can be related to the GPL/FC ratio, the respiratory profile of mitochondria originating from UCP1 KO mice was analyzed using the assay shown in Fig 4B. We could neither determine any systematic DV-dependent differences for basal nor for oligo-induced (= basal uncoupled), G3P-induced, GDP-induced, or (V) Anti A-induced (=non-mitochondrial) respiration (Fig S7F and G). In summary, these data demonstrate that mitochondrial GPL/FC is critical for G3P-dependent respiration and UCP1 function in BAT.

## UCP1 function depends on STARD3-mediated cholesterol import into mitochondria in brown adipocytes

Steroidogenic acute regulatory protein (STARD) 3 facilitates cholesterol import into mitochondria (Elustondo et al, 2017). Thus, we asked if modulation of STARD3, which is expressed at significant but not different levels in murine WAT and BAT (Fig 5A), affects UCP1 activity in primary brown adipocytes. The following mitochondrial bioenergetics profile was determined in intact cells using microplate-based respirometry (Fig 5B) (Schweizer et al, 2019): (1) basal respiration; (2) basal uncoupled respiration after blocking ATP synthase with oligo; (3) induction of UCP1 by addition of isoproterenol (ISO) as beta-adrenergic agonist; (4) assessment of the maximal respiratory capacity by using carbonyl cyanide 4-(trifluoromethoxy) phenylhydrazone (FCCP) as uncoupling agent; (5) non-mitochondrial $O_2$ consumption applying antimycin A (Anti A). UCP1 activity was calculated by subtraction of the OCR determined at (2) (basal uncoupled respiration, Oligo) from the OCR at (3) (UCP1 induction, ISO).

Lentiviral overexpression of *STARD3* (elevating mitochondrial FC contents and lowering GPL/FC) reduced the UCP1-dependent OCR by 1.8-fold (Fig 5C–G). In agreement, its knockdown using RNAi (lowering mitochondrial FC contents and elevating GPL/FC) enhanced UCP1-mediated respiration and activity 2.2-fold (Fig 5H–L). In brown adipocytes originating from UCP1 knockout animals, silencing of *STARD3* did not affect OCR after beta-adrenergic stimulation with ISO (Fig 5M–O), verifying that STARD3-mediated FC transport can be linked to UCP1-dependent uncoupled respiration.

**Figure 4. 1UCP1 function in BAT mitochondria depends on their GPL/FC ratio.**
**(A)** Scheme illustrating principles for fusion of GPL- and FC-containing DV with mitochondria. **(B)** Microplate-based respirometry assay applied to profile UCP1 activity in mitochondria isolated from BAT of mice housed at 23°C. The oxygen consumption rate (OCR) was measured at (1) basal conditions, after injections of (2) oligomycin (to inhibit CV, equals basal leak respiration), (3) G3P (to fuel electron transport chain and UCP1 activity), (4) GDP (to inhibit UCP1 activity), and (5) antimycin A (to block electron transport chain, equals non-mitochondrial respiration). **(C, D, E)** Mitochondrial bioenergetics of BAT WT (C), mixed WT-UCP1 KO (1:1) (D), and UCP1 KO mitochondria. Shown are means ± SD. *$P < 0.05$, indicate a significant difference for the comparisons control versus 100% GPL or 80% GPL-20% FC or 60% GPL-40% FC or 40% GPL-60% FC; determined using a two-sided *t* test. **(E)** loaded with donor vesicles (DV) containing varying ratios in GPL (PC, PA) and FC. Shown are data normalized to the OCR determined before G3P injection (19.0 min). **(B, C, D, F)** UCP1 activity calculated by subtraction of the OCR at (4) (GDP) from (3) (G3P) determined in (B, C, D). **(G, H)** Lipid class composition, (H) PC/FC and GPL/FC of BAT mitochondria that were untreated or loaded with DV containing varying ratios in GPL (POPC, POPA) and FC (n = 3). Measured with mass spectrometry. Shown are means ± SD. *$P < 0.05$, **$P < 0.01$, ***$P < 0.001$ indicate a significant difference for the comparisons control versus 100% GPL, 100% GPL versus 80% GPL-20% FC, 80% GPL-20% FC versus 60% GPL-40% FC and 60% GPL-40% FC versus 40% GPL-60% FC; determined using a two-sided *t* test. **(I, J)** Correlation of UCP1 activity with contents of FC (I) or GPL (J) added to DV. **(K, L)** Correlation of UCP1 activity with contents of FC (K) or GPL (L) measured with mass spectrometry in BAT mitochondria incubated with donor vesicles. $R^2$ indicate Pearson's correlation coefficients. Each mitochondrial sample was isolated from tissue pooled of n = 5 mice. Source data are available for this figure.

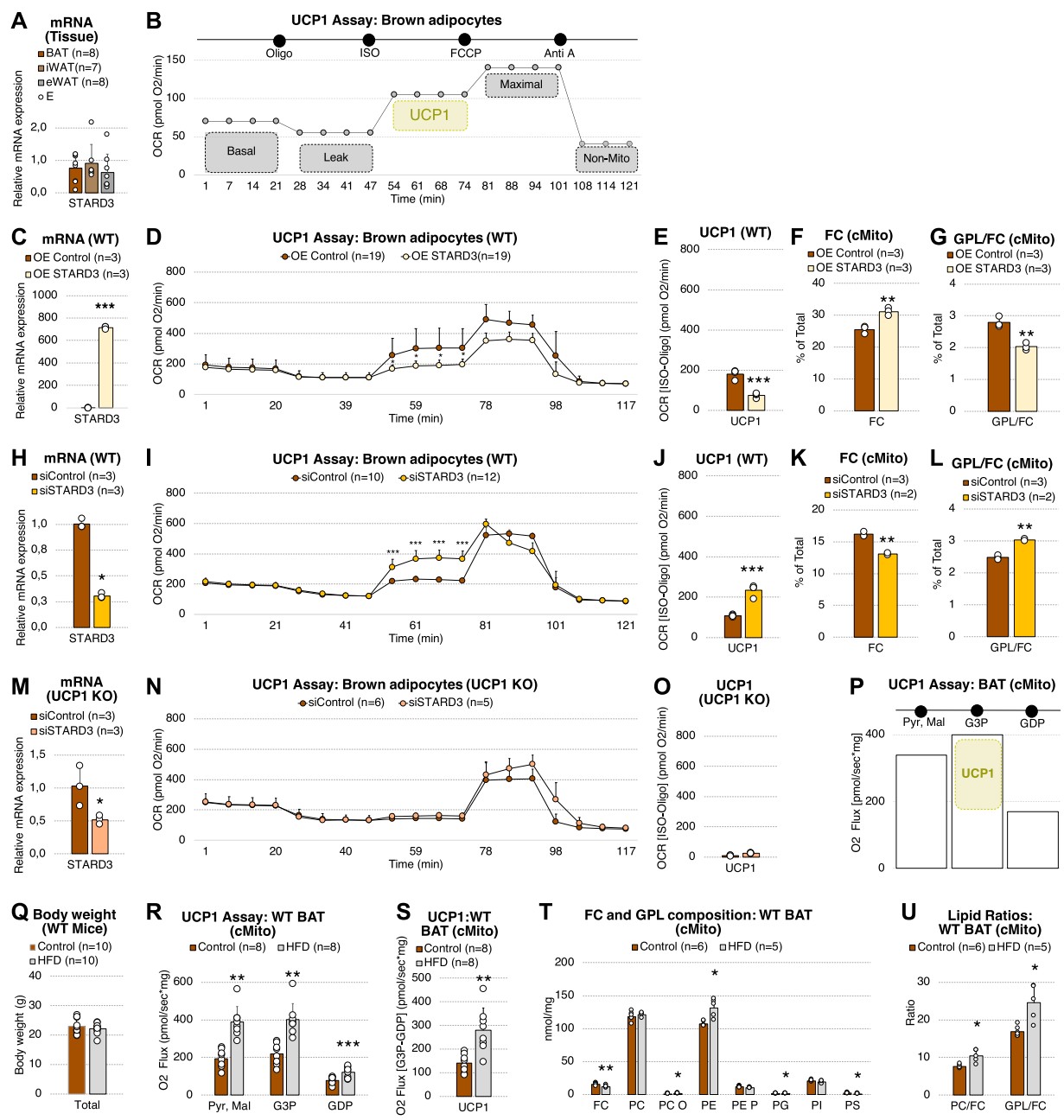

**Figure 5. UCP1 function depends on STARD3-mediated cholesterol import and the dietary fat content.**
**(A)** STARD3 mRNA expression in BAT (n = 8), iWAT (n = 7), and eWAT (n = 8) from WT animals. **(B)** Microplate-based respirometry assay applied to profile UCP1 activity in primary brown adipocytes. The oxygen consumption rate (OCR) was measured at (1) basal conditions, after injections of (2) oligomycin (Complex V inhibition), (3) ISO (UCP1 induction), (4) FCCP (maximal $O_2$ consumption), and (5) antimycin A (inhibition of electron transport chain). **(B, C, D, E, F, G)** STARD3 mRNA expression (n = 3), (D) Mitochondrial bioenergetics (n = 19), measured with the assay described in (B, E) UCP1 activity calculated by subtraction of the OCR at (2) (Oligo) from (3) (ISO), determined in (D) (n = 19), (F) FC (n = 3) and (G), GPL/FC (n = 3) of WT primary brown adipocytes overexpressing (OE) STARD3 and thereof isolated mitochondria. (H) STARD3 mRNA expression (n = 3). **(I)** Mitochondrial bioenergetics (n = 10–12). **(J)** UCP1 activity (n = 10–12). **(K, L)** FC (n = 3) and (L), GPL/FC (n = 3) of WT primary brown adipocytes treated with siRNA against STARD3 and thereof isolated mitochondria. **(M)** STARD3 mRNA expression (n = 3). **(N, O)** Mitochondrial bioenergetics (n = 5–6) and (O), UCP1 activity (n = 5–6) of UCP1 KO primary brown adipocytes treated with siRNA against STARD3. **(P)** High-resolution respirometry assay to profile UCP1 activity in mitochondria isolated from BAT of mice used in the dietary intervention experiments (control or HFD for 14 d). OCR was measured after injections of (1) Pyr and Mal (complex I substrates), (2) G3P (to fuel electron transport chain and UCP1 activity) and (3) GDP (to inhibit UCP1 activity). **(Q)** Body weight (n = 10). **(R)** $O_2$ flux after applying the UCP1 assay described in p (n = 8). **(S)** UCP1 activity calculated by subtraction of the $O_2$ flux at (3) (GDP) from (2) (G3P) determined in r (n = 8). **(T)** Lipid class composition (n = 5–6). **(U)** PC/FC and GPL/FC of mitochondria isolated from BAT of mice fed a control or HFD for 14 d. Shown are means ± SD. *$P$ < 0.05, **$P$ < 0.01, ***$P$ < 0.001 indicate a significant difference determined using a two-sided $t$ test. Each mitochondrial sample was isolated from tissue pooled of n = 5 mice. SDs for panels (D, I, N) are partly too small to be visible. They can be found in the Source Data.
Source data are available for this figure.

## The dietary fat content influences mitochondrial GPL/FC ratio and UCP1 function in BAT of mice

Finally, we asked whether diet influences mitochondrial GPL/FC ratio and UCP1 activity. High-fat feeding has previously been shown to cause an adaptive induction of norepinephrine-induced UCP1-dependent thermogenesis in mice, attenuating body weight gain and obesity, especially in 129S mouse strains having a very high capacity for UCP1-mediated uncoupled respiration (Feldmann et al, 2009; Luijten et al, 2019). Here, respiration in isolated BAT mitochondria was investigated by high-resolution respirometry using the following assay (Fig 5P): After (1) stimulating basal respiration with the complex I substrates pyruvate and malate, (2) the electron transport chain and UCP1 were fueled using G3P and subsequently (3) inhibited by injecting GDP. UCP1 activity was calculated by subtraction of the OCR at (3) (UCP1 inhibition, GDP) from the OCR at (2) (UCP1 fueling, G3P). We found that feeding 129 SV/ev Tac mice a palm oil-based high fat diet (HFD; not containing FC) (Table S3) for 14 d did not alter body weight (Fig 5Q), but significantly increased basal and UCP1-mediated $O_2$ flux almost twofold (Fig 5R and S), as well as GPL levels, i.e. PE and GPL/FC in BAT mitochondria (Fig 5T and U).

# Discussion

Our experiments unveil that the lipidomic organization of mitochondria is critical for brown adipocyte function. Applying an established data analysis strategy (Kindt et al, 2018; Ecker et al, 2021) in combination with an independent approach based on machine learning, we reveal that BAT mitochondria differ most significantly in their PC and FC contents from those of other tissues, including WAT. Biochemical purification of mitochondria was carried out using differential centrifugation. Compared to density gradient centrifugation, this procedure requires substantially less time allowing respiratory measurements and less tissue reducing the numbers of euthanized animals. Proteomic and Western blot analyses confirm a highly enriched mitochondrial fraction containing negligible contaminations with ER and MAM. Mitochondria form contacts with the smooth ER called MAM (also called MERC, mitochondria-ER contact) with a thickness of ~10–50 nm that have important roles in maintenance of lipid and $Ca^{2+}$ homeostasis (Kornmann et al, 2009; Kornmann, 2013; Giacomello & Pellegrini, 2016). Previous studies indicate that biochemical depletion of MAM from ER is more problematic than from mitochondria (Wieckowski et al, 2009; Giacomello & Pellegrini, 2016; Ma et al, 2017).

Until now, mitochondrial cholesterol contents were thought to range around 10% (GPL/FC: 9) (as determined here in the liver and skeletal muscle) (van Meer et al, 2008). Astonishingly, with 2% of FC (equaling GPL/FC: 50) in BAT mitochondria, we found even less. Because expression of SREBP2 target genes relevant for FC biosynthesis (Horton et al, 2002) and STARD3 required for mitochondrial FC import (Elustondo et al, 2017) were similar between BAT and WAT, we conclude that GPL rather than FC metabolism provokes the immense GPL/FC ratio in BAT. PC and PE de novo synthesis capacities are significantly higher in murine BAT compared to WAT. After housing mice at 4°C, 23°C, and 30°C, we found

that GPL levels in mitochondria correlate with adipose tissue browning. In agreement, it was previously reported that cold exposure (4°C) alters GPL metabolism-related gene expression and PC, PE, and CL contents in BAT of mice (Marcher et al, 2015; Lynes et al, 2018). In human serum, levels of polyunsaturated PE and PG species are elevated upon mild cold exposure (14°C) (Lynes et al, 2018).

This study suggests that the ratio of GPL/FC, rather than GPL or FC itself, is critical for thermogenic function. After activation by free FA, UCP1, which is located in the IMM, increases the transport of protons ($H^+$) through the IMM to dissipate the mitochondrial $H^+$ gradient and to convert the energy of substrate oxidation into heat. Details of UCP1-mediated uncoupling still remain elusive. So far, the following models have been proposed: (A) UCP1 functions as uniporter, transporting $H^+$ from outside of the IMM into the mitochondrial matrix, induced by allosteric binding of FA (Rial & Gonzalez-Barroso, 2001; Cannon & Nedergaard, 2004); (B) UCP1 functions as $OH^-$ uniporter, transporting $OH^-$ into the mitochondrial matrix, induced by allosteric binding of FA (Nicholls, 2006); (C) FA bind to the channel of the UCP1 protein providing their carboxylic groups to translocate $H^+$ from outside the IMM into the matrix (the "$H^+$ buffering model") (Klingenberg & Huang, 1999); (D) UCP1 uses FA anions, which cannot dissociate from the protein because of their hydrophobic interactions, to shuttle $H^+$ into the mitochondrial matrix (Fedorenko et al, 2012); (E) UCP1 transports FA anions from the mitochondrial matrix to the outside of the IMM, where they bind $H^+$. After flip-flop of the protonated FA across the lipid bilayer, $H^+$ is released into the matrix (the "FA cycling model") (Garlid et al, 1998). Flip-flop of protonated FA, including palmitate, across lipid bilayers essentially depends on its cholesterol content (Brunaldi et al, 2005). Cholesterol promotes the order in acyl chains of GPL inhibiting their motion in the membrane. Thereby, it increases lipid packing density and the energy needed to create a void that extends into the leaflet of the lipid bilayer to which the FA flops (Kleinfeld et al, 1997). This suggests that a high GPL/FC ratio (low FC content and reduced lipid packing) might increase UCP1 function by facilitating FA cycling across the IMM. In addition, a high GPL/FC ratio might promote FA transport from the cytoplasm through the outer mitochondrial membrane to the intermembrane space.

Because the presence and activity of human BAT correlate with higher energy expenditure, lower adiposity, a reduced risk of insulin resistance and coronary artery disease, induction of UCP1 activity and thermogenesis is considered a promising strategy to combat metabolic and cardiovascular diseases (Chondronikola et al, 2014; Cypess et al, 2015; Iwen et al, 2017; Becher et al, 2021). Targeting the mitochondrial GPL/FC ratio might be useful to trigger BAT activity. We propose that STARD3, rather than cholesterol or GPL de novo synthesis pathways, might be a promising target. Cellular cholesterol and GPL synthesis are highly coordinated and dynamic processes, required for cell differentiation and proper organelle development (Kasturi & Wakil, 1983; Jackowski, 1994; Ecker et al, 2010). Manipulations might interfere with fundamental cellular functions. For example, statin-mediated inhibition of 3-hydroxy-3-methylglutaryl coenzyme A reductase (HMGCR), a key enzyme in cholesterol biosynthesis, prevents adipocyte browning and BAT activity in humans (Balaz et al, 2019). In contrast, manipulation of STARD3 expression alters mitochondrial FC content, as shown

here and by others (Charman et al, 2010; Balboa et al, 2017). It does not regulate cellular cholesterol biosynthesis including *SREBP2* processing and expression of its target genes, such as *HMGCR* and low-density lipoprotein receptor (*LDLR*) (Wilhelm et al, 2017). Interestingly, STARD3 has also been proposed as a molecular target against various tumors, including colorectal, gastric and prostate tumors (Asif et al, 2021). Just recently, a potent STARD3 inhibitor ("VD1") was identified using a pharmacophore-based virtual screening (Lapillo et al, 2019). In our experiments, dietary fat content also influenced the mitochondrial GPL/FC ratio in BAT and UCP1 function. Whether the dietary sterol content impacts the mitochondrial lipidome is unknown, but would be worth to study. In general, the identification of exogenous factors impacting mitochondrial GPL and FC contents might open new ways to promote BAT energy expenditure and prevent or treat associated pathophysiological conditions and diseases.

## Materials and Methods

### Ethics statement

All mouse experiments were performed according to the relevant ethical guidelines. Breeding and experimental use of mice used for this study were approved by the local institution in charge (Regierung von Oberbayern; approval numbers: 55.2-1-54-2531-99-13-2015, 55.2-1-54-2532-17-2015, 55.2-1-54-2532-34-2016, and ROB-55.2-2532.Vet_02-19-193).

### Mouse housing and experiments

If not stated otherwise, all studies were performed in 8–12-wk-old SPF WT 129 SV/ev Tac (also termed 129S6 Sv/EV Tac) mice or UCP1 KO 129S1/SvImJ mice (male and female) (Li et al, 2014) fed a chow diet (V1534, Ssniff), housed at 23°C and 50–60% relative humidity with a 12 h light-dark cycle.

For adipose tissue browning studies, 10-wk-old SPF 129 SV/ev Tac male mice fed a chow diet (V1534, Ssniff) were housed at 4°C, 23°C, and 30°C for 7 d.

For short-term high-fat feeding studies, mice were fed an experimental control or HFD (Table S3) for 14 d.

To quantify de novo PC and PE synthesis via the Kennedy pathway, mice were supplemented simultaneously with choline (D9) and ethanolamine (D4) (each 1 mg/mouse; dissolved in 100 μl 0.9% NaCl solution) via intraperitoneal injection (i.p.) for 2 h, before PC (D9), PC (D4), and PE (D4) were analyzed in BAT, iWAT, and eWAT.

### Cell culture and RNAi

4–6-wk-old male and female mice were used. The stromal vascular fraction was isolated from interscapular BAT and adipocyte precursor cells were cultured as previously described (Schweizer et al, 2019; Oeckl et al, 2020). The medium was changed every other day. At 80-100% confluency, induction medium (10% FBS superior [Biochrom]; 90% DMEM D5796, 1 μM IBMX, 125 μM indomethacin, 250 μM dexamethasone, 850 nM insulin, 1 nM T3 [Sigma-Aldrich];

1:250 pen/strep, 1:250 gentamycin [Biochrom]; 1 μM rosiglitazone [Biomol]) was added. After 48 h, it was replaced with differentiation medium (induction medium excluding IBMX, indomethacin, dexamethasone) for 5 d.

siRNA-mediated gene silencing was conducted using reverse transfection with DsiRNA or negative control DsiRNA (Integrated DNA Technologies) as previously published (Oeckl et al, 2020). Briefly, adipocytes were maintained in differentiation medium containing transfection mix (Opti-MEM [Gibco], 2.5 μl/ml Lipofectamine RNAiMAX [Thermo Fisher Scientific], 50 nM DsiRNA) for 72 h (first 24 h without antibiotics). The following DsiRNA sequences were used: StAR-related lipid transfer domain containing 3 (*STARD3*; 5′-GGAUCAUCGAGCUAAAUACCAACAC-3′; 3′-GACCUAGUAGCUCGAUUUAU GGUUGUG-5′) and non-targeting control (*NTC*; 5′-CGUUAAUCGCGUA UACGCGUAT-3′; 3′-AUACGCGUAUUAUACGCGAUUAACGAC-5′).

### Lentiviral overexpression of STARD3

cDNA was generated from murine liver RNA using the Tetro-cDNA Synthesis Kit (BIO-65042; Meridian Bioscience), before STARD3 was PCR-amplified using the NEB Phusion High Fidelity PCR-Kit with primers containing restriction sites for *Not1* and *BamHI* (forward primer: 5′-TAAGCAGGATCCATGAGCAAGCGACCTGGTGATCT-3′, reverse primer: 5′-TGCTTAGCGGCCGCTCAAGCTCGGGCCCCCAGCT-3′). After STARD3 PCR amplicons were gel-purified (Promega Wizard Sv Gel and PCR Cleanup System), digested with *BamHI-HF* and *Not1-HF*, they were cloned into the lentiviral expression vector pCDH-PGK in a 3:1 ratio (gift from Kazuhiro Oka [#72268; http://n2t.net/addgene: 72268; RRID:Addgene_72268; Addgene plasmid]) using NEB T4-Ligase. pCDH-PGK-STARD3 was transformed into NEB Stable Competent E.Coli (High Efficiency) (C3040H) and purified using the PureYield Plasmid Miniprep System as well as the Nucleobond Xtra Midi Plus (Macherey Nagel).

For production of lentiviral particles, pCDH-PGK-STARD3 or pCDH-PGK (as a control) were co-transfected together with the envelope and packaging plasmids pMD2.G and psPAX2 into HEK293T cells using calcium phosphate transfection at 50% confluency. On two consecutive days after transfection, primary brown pre-adipocytes isolated from BAT were transduced on the first day of differentiation. Therefore, virus-containing medium was collected from transfected HEK293T cells and enriched with polybrene (8 μg/ml) for 24 h to obtain the desired MOI. The lentiviral titer was determined using the One-Wash Lentivirus Titer Kit, HIV-1 p24 ELISA (TR30038; Origene). 72 h post transduction of primary brown adipocytes, mRNA expression and mitochondrial respiration were analyzed using the microplate-based respirometry assay for intact cells.

### Isolation of crude mitochondria from tissue

Mitochondria were enriched using differential centrifugation as previously described (Pallotti & Lenaz, 2007; Cannon & Nedergaard, 2008). Centrifugation steps were performed and buffers were kept at 4°C. Per mitochondrial sample, tissue from five mice was pooled.

BAT was minced in STE buffer (250 mM sucrose, 5 mM Tris, 2 mM EGTA; pH 7.4) containing 2% fatty acid–free BSA, disrupted with a Potter Elvehjem tissue homogenizer (2–3 strokes) and filtered

through 250 $\mu$M nylon gaze. Differential centrifugation was carried out at 8,500$g$, 10 min; 800$g$, 10 min; 10,000$g$, 10 min.

eWAT and iWAT were minced and digested in HBSS containing 1 g/l collagenase (Sigma-Aldrich) and 40 g/l BSA Fraction V (Carl Roth) before being filtered through a 250 $\mu$M nylon gaze. The fraction was washed three times (500$g$, 2 min) with STE containing 4% fatty acid–free BSA to isolate mature adipocytes, which were disrupted using a Potter Elvehjem tissue homogenizer. Differential centrifugation was carried out at 800$g$, 10 min; 10,000$g$, 10 min.

Liver was perfused with 0.9% NaCl minced in STE and disrupted with a Potter Elvehjem tissue homogenizer (five strokes). Differential centrifugation was carried out at 1,000$g$, 3 min; 11,600$g$, 10 min.

Skeletal muscle (musculus biceps femoris) was minced in preparation buffer (0.1 M KCL, 0.05 M Tris–HCL; 2 mM EGTA; pH 7.4), digested in digestion buffer (0.1 M KCL, 0.05 M Tris–HCL, 1 mM ATP, 0.5% fatty acid–free BSA, 245.7 units/100 ml Protease Type VIII [Sigma-Aldrich]; 2 mM EGTA, 5 mM MgCl$_2$; pH 7.4) and homogenized using a Polytron homogenizer. Differential Centrifugation was carried out at 490$g$, 10 min; 10,400$g$, 10 min; 10,400$g$, 10 min; 3,800$g$, 10 min.

For high-resolution respirometry and microplate-based respirometry analyses, mitochondrial pellets were transferred to STE containing 0.4% essentially fatty acid-free BSA, and for lipidomic analyses to 0.1% SDS.

## Preparation of ER, MAM, and pure mitochondria from tissue

ER, MAM and pMito were prepared using density gradient centrifugation as previously described by Wieckowski and colleagues (Wieckowski et al, 2009) from BAT or liver pooled from 15 mice.

## Isolation of mitochondria from primary brown adipocytes

Mitochondria were isolated from primary brown adipocytes using a commercially available mitochondria isolation kit (130-096-946; Miltenyi Biotech) according to the manufacturer's instructions. Briefly, cells were harvested and washed three times with ice-cold PBS (300$g$, 10 min, 4°C) before lysis and homogenization using a G18 needle. After magnetic labeling with mouse anti-TOM22 Microbeads and enrichment using a LS column in the magnetic field of a QuadroMACS separator, mitochondria were pelleted and washed with both, STE + 2% and 0.4% essentially FA-free BSA (13,000$g$, 2 min, 4°C). Per mitochondrial sample, cells from BAT of three mice were pooled. For lipidomic analyses, mitochondria were transferred into 0.1% SDS.

## Preparation of donor vesicles and their fusion with BAT mitochondria

Small unilamellar donor vesicles were generated according to the principles described by Kainu and colleagues (Kainu et al, 2010), and Suresh and London (Suresh & London, 2022). Briefly, palmitoyl-oleoyl-phosphatidylcholine (POPC, Avanti Polar Lipids), palmitoyl-oleoyl-phosphatidic acid (POPA, Avanti Polar Lipids) and free cholesterol (FC, Sigma-Aldrich) were mixed in varying ratios (0% FC, 95% POPC, 5% POPA; 20% FC, 75% POPC, 5% POPA; 40% FC, 55% POPC,

5% POPA and 60% FC, 35% POPC, 5% POPA) in chloroform/methanol (9:1) at a total lipid concentration of 0.5 mM. The solvent was evaporated using a nitrogen stream, leaving a thin lipid film on Pyrex tube walls. After PBS was added, donor vesicles were generated using sonication and vortexing intervals of 6 × 5 min and 7 × 1 min, respectively. Undispersed lipids and probe particles were removed (3,000$g$, 5 min). For fusion with mitochondria, donor vesicle solutions were incubated with 10 mM methyl-$\alpha$-cyclodextrin (AraChem Cyclodextrin-shop) for 1 h at 63°C and 300 rpm (Eppendorf Thermomixer Compact) and subsequently with 0.5–0.8 mg of isolated BAT mitochondria for 1 h at 4°C. Next, mitochondria-vesicle solutions were centrifuged at 10,000$g$ and 4°C for 10 min. Control mitochondria were treated as those incubated with donor vesicles.

Contamination with non-incorporated vesicles is highly unlikely because the "mitochondria-vesicle" mixtures (in PBS) were centrifuged after incubation at 10,000$g$ for 10 min. DV not incorporated into mitochondria should be recovered in the supernatant because the small, sonicated unilamellar vesicles (<100 nm) generated by sonication should not sediment at 10,000$g$. This would require ultracentrifugation (120,000$g$, 30 min) as shown by Tortorella and London (Tortorella & London, 1994).

## RNA isolation and quantitative real-time PCR analysis

Total RNA was extracted from total tissue or cells using the RNEasy Mini Kit (QIAGEN). The purity and integrity of the RNA were assessed using the Agilent 2100 bioanalyzer (Agilent Technologies). For real-time PCR, 1 $\mu$g RNA was transcribed into cDNA using the Reverse Transcription System from Promega. Real-time quantitative RT–PCR analysis was performed using the Light Cycler LC 480 (Roche). The following primer were used: StAR related lipid transfer domain containing 3 (*STARD3*; fw: 5'-CACCTTCTGCCTCTTCGTCACC-3'; rev: 5'-AACACTGGCATCCGGAAGAA-3'), Acetyl-CoA acetyltransferase 2 (*ACAT2*; fw: 5'-GCAGAGGGCCAAGGTGGCT-3'; rev: 5'-TGCACCCACACTGG CTTGTCG-3'), Cytochrome P450 family 51 subfamily A member 1 (*CYP51A1*; fw: 5'-CCAATTCCATTCCTTGGCCATGC-3'; rev: 5'-CTTCTTCTGC ATTCAGGTCTTCG-3'), 7-dehydrocholesterol reductase (*DHCR7*; fw: 5'-GGCCATGCTAGTCTGGCAGA-3'; rev: 5'-ACCTGGCAGAAATCTGTGGCA G-3'), Farnesyl-diphosphate farnesyltransferase 1 (*FDFT1*; fw: 5'-CCGACAAGTGCTGGAGGACTT-3'; rev: 5'-TGGCAGATGTCATCGATCACTG-3'), Geranylgeranyl diphosphate synthase 1 (*GGPS1*; fw: 5'-CACAGG-CATTTAATCACTGGCTG-3'; rev: 5'-CTGGGAAACCACGTCGGAGC-3'), 3-hydroxy-3-methylglutaryl-coenzyme A reductase 1 (*HMGCR1*; fw: 5'-ATGCCTTGTGATTGGAGTTGG-3'; rev: 5'-TGGACGACCCTCACGGCTTTC-3'), 3-hydroxy-3-methylglutaryl-coenzyme A synthase 1 (*HMGCS*; fw: 5'-GTCTCCTTGCTTTGCTCGTT-3'; rev: 5'-TCCAGCATCTACACCATCGT-3'), Isopentenyl-diphosphate delta isomerase 1 (*IDI1*; fw: 5'-GGA-GAGGATTGAAGTACAGCTCT-3'; rev: 5'-CAATAAGAATACACATCTCCGCTAG-3'), Low density lipoprotein receptor (*LDLr*; fw: 5'-CTAGCGATGCATTTTCCGTC-' rev: 5'-GTCATCGCCCTGCTCCTT-3'), Lanosterol synthase (*LSS*; fw: 5'-GGCAGAGATGGACTTATTATCAAGC-3'; rev: 5'-TCAGCCTGCAGCTTGGCAT-3'), Mevalonate diphosphate decarboxylase (*MVD*; fw: 5'-CTCAGCCG-CAGGCTATGC-3'; rev: 5'-AGACTGCGGCACGCACTG-3'), Mevalonate kinase (*MVK*; fw: 5'-ATATCCCTGGAGTGTGAGCG-3'; rev: 5'-CCACTGTGGCTTGCTCTAGA-3'), Phosphomevalonate kinase (*PMVK*; fw: 5'-TGGATGCGAGCACCTACAAGG-

3'; rev: 5'-TGGATGCGAGCACCTACAAGG-3'), Sterol-C5-desaturase-like (*SC5DL*; fw: 5'-GGCCTGCACCACAGACTGGT-3'; rev: 5'-CTCTGAAG-GAAGCCGTCCACA-3'), Squalene epoxidase (*SQLE*; fw: 5'-CCGGAAGT-GATCATCGTCGGGGAT-3'; rev: 5'-GACTCATGATGAATCGGCCATGG-3'), Sterol regulatory element binding transcription factor 2 (*SREBF2*; fw: 5'-GACGTTCAGCACCGCTCC-3'; rev: 5'-AGGCTTTGCACTTGAGGCTG-3'), Glyceraldehyde 3-phosphate dehydrogenase (*GAPDH*; fw: 5'-CGCCTGGAGAAACCTGCC-3'; rev: 5'-AGCCGTATTCATTGTCATACCAGG-3'), 18S ribosomal RNA (*18S*; fw: 5'-GGCCCTGTAATTGGAATGAGTC-3'; rev: 5'-CCAAGATCCAACTACGAGCTT-3'), ß-Actin (*ß-Act*; fw: 5'-CTCTGGCTCCTAGCACCATGAAGA-3'; rev: 5'-GTAAAACGCAGCTCAGTAA-CAGTCCG-3'), Hypoxanthine-guanine phosphoribosyl transferase (*HPRT1*; fw: 5'-CGATGATGAACCAGGTTATGA-3'; rev: 5'-TCCTTCATGA-CATCTCGAGCAAGTC-3'). *18S*, *ß-Act*, *GADPH* and *HPRT1* were used as reference genes. Relative quantification was carried out using LightCycler 480 Software 1.5.1 (Roche) and the $2^{-\Delta\Delta C_T}$ method.

## Protein isolation and Western blotting

Protein from eWAT, iWAT, and BAT was isolated and homogenized in 5 $\mu$l/mg RIPA buffer (150 nM NaCl, 50 mM Tris-base, 1% NP-40, 0.25% Na-desoxycholate, 1 mM EDTA, 0.1% protease inhibitor, 0.1% phosphatase inhibitor; Sigma-Aldrich) using a dispersing device (Miccra D-1; Miccra UHS-RS Technology). The homogenate was centrifuged (16,000*g*, 15 min, 4°C) and the protein concentration of the clear layer was quantified according to the manufacturer's instructions using the Pierce BCA Protein Assay Kit (Thermo Fisher Scientific). 20–100 $\mu$g of protein was separated using SDS–PAGE (12.5% gel), transferred on a nitrocellulose membrane and incubated with Na$^+$/K$^+$-ATPase $\alpha$1 C464.6 (sc-21712; Santa Cruz) at 1:100, IP3R3 I/II/III B-2 (sc-377518; Santa Cruz) at 1:100, calreticulin (ab92516-1001; Abcam) at 1:1,000, FACL4 (ab155282; Abcam) at 1:200 and cytochrome C A-8 (sc-13156; Santa Cruz) at 1:500 for 1 h. For detection, IRDye 680RD-conjugated anti-mouse or anti-rabbit IgG (LI-COR Biosciences) was used at 1:10,000 for 1 h, visualization was carried out using an Azure Sapphire biomolecular imager (Azure biosystems).

## Proteomics

### Lysis, protein digestion, and peptide fractionation
Isolated organelles were resuspended in lysis buffer consisting of 8 M urea in 50 mM Tris–HCl pH 8, and containing EDTA-free protease inhibitors cocktail (Roche). Lysis was performed by sonication in a Bioruptor Pico (Diagenode) using a 10 cycles program (30 s ON, 30 s OFF), and lysate was subsequently cleared by centrifugation for 10 min at 20,000*g* and 4°C. Protein were reduced with 10 mM DTT at 37°C for 40 min on a thermoshaker at 700 rpm (Eppendorf Thermomixer Compact) and alkylated with 55 mM chloroacetamide at room temperature for 30 min in the dark. Tryptic digestion was performed overnight at 37°C with sequencing grade modified trypsin (1:50 enzyme-to-substrate ratio; Promega) after fourfold dilution with 50 mM Tris–HCl, pH 8. Digests were acidified by addition of formic acid (FA) to 1% (vol/vol) and desalted using Sep-Pak C18 cartridges (Waters). Desalted peptides were resuspended in 10 mM ammonium acetate, pH 4.7, and subjected to peptide fractionation via trimodal mixed mode chromatography on a Dionex Ultra 3000 HPLC system operating an Acclaim Trinity P1 3 $\mu$m

2.1 × 150 mm column (Thermo Fisher Scientific), as previously described (Yu et al, 2017). A total of 32 fractions were collected, vacuum-dried in a SpeedVac (UniEquip), and stored at –20°C until LC–MS/MS analysis.

### LC–MS/MS analysis
Samples were analyzed on a micro-flow LC–MS/MS system using a modified Vanquish pump coupled to an Orbitrap Funsion Lumos mass spectrometer (both Thermo Fisher Scientific). Chromatographic separation was performed via direct sample injection onto the head of a 15 cm Acclaim PepMap 100 C18 column (2 $\mu$m particle size, 1 mm ID; Thermo Fisher Scientific) at a flow rate of 50 $\mu$l/min (Bian et al, 2020). Solvent A was 0.1% FA, 3% DMSO in water, and solvent B was 0.1% FA, 3% DMSO in ACN. Samples were separated with a linear gradient of 3–28% B in 15 min. Total analysis time was 17 min. The Fusion Lumos was operated in positive ion mode, using an electrospray voltage of 3.5 kV, capillary temperature of 325°C, and vaporizer temperature of 125°C. The flow rates of sheath gas, aux gas and sweep gas were set to 32, 5, and 0, respectively. The Fusion Lumos was operated in a data-dependent acquisition to automatically switch between MS and MS/MS. Survey full-scan MS spectra were recorded in the orbitrap at a resolution of 60,000 at m/z 200 and an AGC target value of 4 × 10$^5$ with a maximum injection time (IT) of 50 ms. The MS1 mass range was set to 360–1,300. The isolation width was set to 0.4 m/z, and the first mass was fixed at 100 m/z. The normalized collision energy was set to 32%. Peptide match was set to "preferred," and isotope exclusion was enabled. MS2 spectra for peptide identification were recorded in the ion trap in rapid scan mode with a top speed approach using a 0.6-s duration (isolation window 0.4 m/z, AGC target value of 1 × 10$^4$, maxIT of 10 ms). MS1 and MS2 spectra were acquired in profile and centroid mode, respectively. The dynamic exclusion value was set to 12 s. Measurements were taken from distinct samples.

### Data processing and bioinformatic analysis
Raw data files were processed with MaxQuant (version 1.6.10.43) using default parameters (Cox & Mann, 2008). Spectra were searched against the UniProtKB database (*Mus musculus*, UP000000589, 55,431 entries downloaded on 12.2019). Enzyme specificity was set to trypsin and up to two missed cleavages were allowed. Cysteine carbamidomethylation was set as a fixed modification, whereas Ntem-acetylation of protein and oxidation of methionine were selected as variable modifications. Precursor tolerance was set to 5 ppm, and fragmentation tolerance to 20 ppm. Results were adjusted to a 1% false discovery rate at protein, peptide levels. Identifications were filtered to remove contaminants and decoy hits using in Perseus (v. 1.6.1.1) (Tyanova et al, 2016) prior to subsequent analysis.

The mass spectrometry proteomics data have been deposited in the ProteomeXchange Consortium (Vizcaino et al, 2013) via the PRIDE partner repository with the dataset identifier PXD028128.

## Hierarchical clustering of proteomic data

Proteomics data (Log$_{10}$ [signal intensities]) were hierarchically clustered using a distance matrix based on Euclidean distance and

complete linkage in KNIME 4.3.2. Analytics Platform (https://www.knime.com).

## Lipidomics

Lipid species were annotated according to the proposal for shorthand notation of lipid structures that are derived from mass spectrometry (Liebisch et al, 2020). Measurements were taken from distinct samples. The core lipidomics data from Fig 2 can be found as Supplemental Data 1.

### Internal standards

The following lipid species were applied as internal standards: CE 17:0, CE 22:0, CL 14:0/14:0/14:0/14:0, Cer 18:1; O2/14:0, Cer 18:1; O2/17:0, DG 14:0/14:0/0:0, DG 20:0/20:0/0:0, (D7)FC, HexCer 18:1; O2/12:0, HexCer 18:1; O2/17:0, LPC 13:0/0:0, LPC 19:0/0:0, LPE 13:0/0:0, PC 14:0/14:0, PC 22:0/22:0, PE 14:0/14:0, PE 20:0/20:0, PI 17:0/17:0, PS 14:0/14:0, PS 20:0/20:0, SM 18:1; O2/12:0, TG 17:0/17:0/17:0, and TG 19:0/19:0/19:0.

### Tissue homogenization

Adipose tissue was homogenized with a Precellys 24 tissue homogenizer from Bertin Instruments. Therefore, an amount of ~25–75 mg was added to a Precellys cup containing ceramic beads (V = 2 ml). $H_2O$/MeOH = 1/1 (vol/vol) was added to suspend the samples at a concentration of 0.05 mg/$\mu$l. The homogenizer was operated at 5,000 rpm (Bertin Technologies, Precellys 24 Homogenizer), two cycles of 15 s run time, and a 60 s break interval between both cycles.

### Lipid extraction

Samples were spiked with internal standards before lipid extraction (solvent of standards was removed by vacuum centrifugation). Mitochondria homogenates containing ~100 $\mu$g of protein were extracted according to the protocol described by Bligh and Dyer (Bligh & Dyer, 1959) with a total chloroform volume of 2 ml. A volume of 0.8 ml (for FIA-MS/MS), 0.5 ml (for FIA-FTMS) and 0.3 ml (for FIA-FTMS, CL determination) of the separated chloroform phase was transferred by a pipetting robot (Tecan Genesis RSP 150) and vacuum dried. The residues were dissolved in 0.8 ml methanol/chloroform (3:1, vol/vol) with 7.5 mM ammonium acetate, 1.2 ml chloroform/methanol/2-propanol (1:2:4 vol/vol/v) with 7.5 mM ammonium formate, or 0.5 ml methanol/chloroform (5:1, vol/vol) with 0.005% methylamine, respectively.

For analysis of adipose tissue, homogenates containing a wet weight of 2 mg were extracted instead. A total chloroform volume of 1.6 ml was vacuum dried, dissolved in 1.6 ml methanol/chloroform (3:1, vol/vol) with 7.5 mM ammonium acetate and analyzed by FIA-MS/MS.

### Lipid analysis by mass spectrometry

The analysis of lipids was performed by direct flow injection analysis (FIA), using either a triple quadrupole mass spectrometer (QQQ; FIA-MS/MS) or a hybrid quadrupole-Orbitrap high-resolution mass spectrometer (FIA-FTMS). Both instruments were equipped with a heated electrospray ionization source. FIA-MS/MS (QQQ) was

performed in positive ion mode using the analytical setup and strategy described previously (Liebisch et al, 2004). For the FIA-MS/MS analyses a fragment ion of $m/z$ 184 was used for PC and $m/z$ 193 for choline labeled PC (D9). The following neutral losses were applied: PE, 141; PE (D4), 145; PG, 189; PI, 277, and PS, 185. PE-based plasmalogens (PE P) were analyzed according to the principles described by Berry and colleagues (Zemski Berry & Murphy, 2004). Sphingosine-based Cer were analyzed using a fragment ion of $m/z$ 264. Correction for isotopic overlap of lipid species was performed for all lipid classes.

A detailed description of the FIA-FTMS method was published recently (Horing et al, 2021). TG, DG, and CE were recorded in positive ion mode as $[M+NH_4]^+$ in $m/z$ range 500–1,000 and a target resolution of 140,000 (at $m/z$ 200). CE species were corrected for their species-specific response (Horing et al, 2019). PC, ether PC (PC O), and SM were analyzed as $[M+HCOO]^-$ in negative ion mode in $m/z$ range 520–960 at the same resolution setting. Multiplexed acquisition (MSX) was applied for the $[M+NH_4]^+$ of FC and the corresponding internal standard (D7)FC (Horing et al, 2019). CL were determined as deprotonated adducts in negative ion mode in $m/z$ range 1,200–1,600 from the extract dissolved in methanol/chloroform (5:1, vol/vol) with 0.005% methylamine. Quantification was achieved by multiplication of the spiked-in IS amount with the analyte-to-IS intensity ratio.

## Statistical analysis of lipidomic data

Lipidomic data were analyzed as described previously (Kindt et al, 2018). If measured in different batches, lipid species concentrations were quotient normalized (Dieterle et al, 2006). For generation of volcano plots, all data were $\log_2$ transformed to ensure that they were normally distributed. Lipid species were excluded if they were undetectable in more than 50% of the samples per group. A standard two-sided, unpaired $t$ test assuming unequal variances was used to test for significantly different abundances in the conditions. The Benjamini–Hochberg method to calculate the false discovery rate (FDR) was used to account for multiple testing ($P_{adj}$ < 0.01). Fold changes were calculated as the difference between mean of the log-transferred values in BAT and liver or eWAT or muscle.

## Respiratory function and UCP1 activity analyses in intact cells using microplate-based respirometry

After siRNA-mediated gene silencing or lentiviral overexpression, mitochondrial bioenergetics were assayed using microplate-based respirometry as previously published (Schweizer et al, 2019; Oeckl et al, 2020). Equal amounts of primary brown pre-adipocytes were seeded onto each well of a 96-well Seahorse plate and differentiated into adipocytes as previously described. The cellular OCR of brown adipocytes was measured at 37°C using an XF96 Extracellular Flux Analyzer (Agilent Technologies). Cells were washed twice with respiration medium (DMEM base D5030, 15 mg/liter phenol red [Sigma-Aldrich]; 2 mM GlutaMAX [Life Technologies]; 25 mM Glucose, 31 mM NaCl [Carl Roth]). Prior to measurement, respiration medium was supplemented with 2% essentially fatty acid-free BSA (Sigma-Aldrich) and cells were incubated for 1 h in a 37°C incubator w/o $CO_2$. After measuring the basal respiration of adipocytes, ATP-linked respiration

(5 $\mu$M oligomycin; Sigma-Aldrich), UCP1-mediated uncoupled respiration (0.5 $\mu$M ISO; Sigma-Aldrich) and maximal oxidative capacity (7 $\mu$M FCCP; Sigma-Aldrich) were assessed. To distinguish mitochondrial oxygen consumption from non-mitochondrial oxygen consumption, 5 $\mu$M antimycin A (Sigma-Aldrich) was injected at the end of the measurement. To compare UCP1 activity between different groups, data were normalized to the end point of ATP-linked respiration (oligomycin).

### UCP1 activity analyses in isolated mitochondria incubated with donor vesicles using microplate-based respirometry

Mitochondrial bioenergetics of isolated mitochondria incubated with donor vesicles was assayed using microplate-based respirometry as previously published (Sustarsic et al, 2018). The OCR of mitochondria was measured at 37°C using an XFe96 Extracellular Flux Analyzer (Agilent Technologies). Mitochondrial pellets were resuspended in respiration buffer (125 mM sucrose, 20 mM K$^+$-TES, 1 mM EDTA, 10 mM K$^+$-pyruvate, 2 mM malate [Sigma-Aldrich], 2 mM MgCl$_2$ * 6 H$_2$O, 4 mM KH$_2$PO$_4$ [Carl Roth]; pH 7.2) containing 0.4% essentially fatty acid-free BSA (Sigma-Aldrich) and seeded onto a pre-cooled 96-well Seahorse plate (4 $\mu$g mitochondrial protein/well, protein concentration was determined using the Biuret assay). Mitochondria were pelleted at 2,000$g$ for 20 min at 4°C using a swing bucket centrifuge with fast acceleration and slow break settings. The OCR was measured using two assays. Assay I (Fig 4A): After two basal measurements, ATP-linked respiration (injection of 5 $\mu$M oligomycin) was assessed and general respiration was fueled using glycerol-3-phosphate (G3P) (injection of 5 mM G3P). Assay II (Fig S7A): After three basal measurements, general respiration was fueled using glycerol-3-phosphate (G3P) (injection of 5 mM G3P). In both assays, UCP1 activity was calculated by subtraction of O$_2$ consumed after 3 mM GDP injection (Assay I: OCRs at 42.7 and 48.6 min; Assay II: OCRs at 36.7, 42.6, and 48.5 min) from O$_2$ consumed directly before GDP injection (Assay I: OCR at 42.6 min; Assay II: OCR at 36.6 min). For analysis of non-mitochondrial O$_2$ consumption, in both assays, the electron flow was blocked using a complex-III inhibitor (injection of 5 $\mu$M antimycin A) at the end of each measurement (all compounds Sigma-Aldrich). For a better overview, data were normalized to the OCR determined at 19.0 min (before G3P injection) in Figs 4B–D and S7B. This may lead to negative OCRs after injection with Anti A for certain conditions.

### Analysis of maximal respiratory capacity of isolated mitochondria from BAT incubated with donor vesicles using microplate-based respirometry

Mitochondrial pellets were resuspended in the identical respiration buffer, seeded into a 96-well Seahorse plate and pelleted as for the UCP1 activity measurements described above. General respiration was fueled by injection of 5 mM G3P, UCP1 inhibited with 3 mM GDP and maximal respiration was induced with 7.5 $\mu$M FCCP before the OCRs were determined using a XFe96 Extracellular Flux Analyzer.

### UCP1 activity analyses in isolated mitochondria from BAT of mice fed a HFD using high-resolution respirometry

Respiratory measurements of isolated mitochondria were conducted with the Oroboros O2K respirometer (Oroboros Instruments). Chambers were air-saturated with 2 ml of 37°C pre-heated respiration buffer (KHE; 50 mM KCL, 5 mM TES, 1 mM EGTA, 0.4% essentially fatty acid-free BSA (Sigma-Aldrich); 2 mM MgCl$_2$ * 6 H$_2$O, 4 mM KH$_2$PO$_4$ [Carl Roth]; pH 7.2). To avoid O$_2$ dependent artefacts, the respiration buffer was stirred continuously and oxygen concentration was kept above 50 $\mu$M. After mitochondrial isolation, 0.1 mg/ml mitochondria were transferred into O2K chambers. Protein concentration was determined using the Biuret assay. When basal oxygen consumption was stable, the following compounds were applied sequentially:

5 mM pyruvate (Pyr), 2 mM malate (Mal, C I substrates); 5 mM glycerol-3-phosphate (G3P); 2.5 mM guanosine diphosphate (GDP) as UCP1 inhibitor. To subtract non-mitochondrial oxygen consumption from mitochondrial oxygen consumption, the electron flow was blocked with 2.5 $\mu$M C III inhibitor antimycin A (anti A) at the end of each measurement (all compounds Sigma-Aldrich). The measured OCRs are given related to mitochondrial protein.

### Analysis of maximal respiratory capacity of isolated mitochondria from BAT incubated with donor vesicles using high-resolution respirometry

Respiratory measurements of isolated mitochondria were performed with the Oroboros O2K respirometer (Oroboros Instruments) according to the principles described above. 5 mM pyruvate (Pyr), 2 mM malate (Mal), 5 mM glycerol-3-phosphate (G3P), 2.5 mM GDP as UCP1 inhibitor, and 5 $\mu$M FCCP to induce maximal respiration were applied before O$_2$-flux measurements. Our 3x2-chamber configuration only allows similtaneous comparison of maximal three conditions with two replicates.

### Classification and prediction model

The prediction model was built with KNIME 4.3.2. Analytics Platform (https://www.knime.com). We applied random forest RF (Breiman, 2001), a supervised decision tree learning procedure, to identify most predictive lipid species indicating the sample origin. Only lipid species from GPL and FC detected in all applied tissues were considered including 94 species. We used data obtained from the experiments shown in Fig 2 for model training (BAT, n = 6; liver, n = 3; eWAT, n = 3; muscle, n = 3; $\sum$ n = 15) and verification (liver, n = 11; BAT, n = 7; eWAT n = 6; $\sum$ n = 24). For feature selection and model learning, the training data was 90% portioned and randomly drawn, the information gain ratio was used as tree split criterion, and the number of decision trees to be learned was set to 100 using a static random seed. Application data were obtained from a new experiment comprising mitochondrial samples from mice housed at 4°C, 23°C and 30°C (BAT, iWAT and eWAT; 4°C, 23°C, 30°C; n = 3/condition; $\sum$ n = 27). Classification probabilities (shown in Fig 3C–F) indicate the individual prediction confidences for each tissue class (BAT, eWAT, liver, muscle).

## Code availability

Computer codes can be made available from the corresponding author on request.

# Data Availability

The mass spectrometry proteomics data have been deposited in the ProteomeXchange Consortium (Vizcaino JA et al, 2013) via the PRIDE partner repository with the dataset identifier PXD028128. Lipidomics data can be found as Supplemental Data 1, for details, see Lipidomics section above.

## Ethics & inclusion statement

We aimed to follow principles set out in the Global Code of Conduct. Our research included local researchers throughout the research process. When possible, it was determined in collaboration with local partners.

# Supplementary Information

# Acknowledgements

This work was funded by Deutsche Forschungsgemeinschaft (DFG) grants 395357507- SFB 1371 (J Ecker, M Klingenspor, K-P Janssen), 446175916 - EC 453/4-1 (J Ecker), LI 923/9-1 (G Liebisch), KL 973/14-1 (M Klingenspor). We thank Doreen Mueller and Daniela Kolmeder for excellent technical assistance, Percy A. Knolle and Michael Dudek for supporting the microplate-based respirometry analyses, and Pentti Somerharju for sharing his knowledge in the preparation and application of small unilamellar donor vesicles.

## Author Contributions

S Brunner: data curation, validation, investigation, visualization, methodology, and writing—original draft, review, and editing.
M Höring: data curation, validation, investigation, methodology, and writing—review and editing.
G Liebisch: resources, funding acquisition, validation, investigation, methodology, and writing—review and editing.
S Schweizer: data curation, investigation, and methodology.
J Scheiber: investigation, visualization, methodology, and writing—review and editing.
P Giansanti: investigation, methodology, and writing—review and editing.
M Hidrobo: investigation.
S Hermeling: investigation.
J Oeckl: investigation.
N Prudente de Mello: investigation and methodology.
F Perocchi: investigation.
C Seeliger: investigation.
A Strohmeyer: investigation.
M Klingenspor: resources and investigation.
J Plagge: investigation.
B Küster: investigation.
R Burkhardt: investigation.
KP Janssen: investigation.
J Ecker: conceptualization, resources, data curation, supervision, funding acquisition, validation, investigation, visualization, methodology, project administration, and writing—original draft, review, and editing.

## Conflict of Interest Statement

The authors declare that they have no conflict of interest.

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
