## [Reviewer comments · Life Science Alliance]

Mitochondrial lipidomes are tissue specific - low cholesterol contents relate to UCP1 activity

Sarah Brunner, Marcus Höring, Gerhard Liebisch, Sabine Schweizer, Josef Scheiber, Piero Giansanti, Maria Hidrobo, Sven Hermeling, Josef Oeckl, Natalia Prudente de Mello, Fabiana Perocchi, Claudine Seeliger, Akim Strohmeyer, Martin Klingenspor, Johannes Plagge, Bernhard Küster, Ralph Burkhardt, Klaus Peter Janssen, and Josef Ecker

DOI: <https://doi.org/10.26508/lsa.202402828>

Corresponding author(s): Josef Ecker, University of Regensburg

Review Timeline:

Submission Date:	2024-05-17
Editorial Decision:	2024-05-20
Revision Received:	2024-05-22
Editorial Decision:	2024-05-22
Revision Received:	2024-05-24
Accepted:	2024-05-27

Transaction Report:

Please note that the manuscript was previously reviewed at another journal and the reports were taken into account in the decision-making process at *Life Science Alliance*. Since the original reviews are not subject to Life Science Alliance's transparent review process policy, the reports and author response cannot be published.

May 20, 2024

Re: Life Science Alliance manuscript #LSA-2024-02828-T

Josef Ecker
University of Regensburg
Franz-Josef-Strauß-Allee 11
Regensburg 93053
Germany

Dear Dr. Ecker,

Thank you for submitting your manuscript entitled "Mitochondrial lipidomes are tissue specific - a high phospholipid to cholesterol ratio relates to UCP1 activity" to Life Science Alliance. We invite you to submit a revised manuscript addressing the remaining reviewers comments.

Thank you for this interesting contribution to Life Science Alliance. We are looking forward to receiving your revised manuscript.

Sincerely,

B. MANUSCRIPT ORGANIZATION AND FORMATTING:

May 22, 2024

RE: Life Science Alliance Manuscript #LSA-2024-02828-TR

Prof. Josef Ecker
University of Regensburg
Franz-Josef-Strauß-Allee 11
Regensburg 93053
Germany

Dear Dr. Ecker,

Thank you for submitting your revised manuscript entitled "Mitochondrial lipidomes are tissue specific - low cholesterol contents relate to UCP1 activity". We would be happy to publish your paper in Life Science Alliance pending final revisions necessary to meet our formatting guidelines.

- please be sure that the authorship listing and order is correct
- please add your main, supplementary figure, and table legends to the main manuscript text after the References section
- please add the Twitter handle of your host institute/organization as well as your own or/and one of the authors in our system
- please note that the titles in the system and manuscript file must match
- please use the [10 author names et al.] format in your references (i.e., limit the author names to the first 10)
- please label your supplementary tables as Table S1, Table S2, etc.
- please add callouts for Figures 3L and P; 4G; S1F and M; S2A-L; S3A-J and S4A-J to your main manuscript text
- please repeat the PRIDE accession information in the Data Availability statement. This should also be made publicly accessible at this point, removing the need to include Reviewer login info.

FIGURE CHECKS:

- please incorporate the References listed under Table S1 into the main References list

A. FINAL FILES:

B. MANUSCRIPT ORGANIZATION AND FORMATTING:

Sincerely,

May 27, 2024

RE: Life Science Alliance Manuscript #LSA-2024-02828-TRR

Prof. Josef Ecker
University of Regensburg
Franz-Josef-Strauß-Allee 11
Regensburg 93053
Germany

Dear Dr. Ecker,

Thank you for submitting your Research Article entitled "Mitochondrial lipidomes are tissue specific - low cholesterol contents relate to UCP1 activity". It is a pleasure to let you know that your manuscript is now accepted for publication in Life Science Alliance. Congratulations on this interesting work.

DISTRIBUTION OF MATERIALS:

Again, congratulations on a very nice paper. I hope you found the review process to be constructive and are pleased with how the manuscript was handled editorially. We look forward to future exciting submissions from your lab.

Sincerely,
